# Engagement and retention in HIV care in rural Southern U.S. using an All-Payer Claims Database

Jun Tao[1]*, Mofan Gu[2], S. Alexandra Marshall[2], Jure Baloh[2], Mike Cima[3], Cornelius Mabin[4], Nickolas D. Zaller[2], Dan Moore[5], Hannah Parent[1], Heather Horton[2], Ira B. Wilson[6], Amy S. Nunn[7], Philip A. Chan[1]

1 Department of Medicine, Brown University, Providence, Rhode Island, United States of America, 2 Fay W. Boozman, College of Public Health, University of Arkansas for Medical Sciences, Little Rock, Arkansas, United States of America, 3 Department of Health Services, Policy & Practice, Brown University School of Public Health, Providence, Rhode Island, United States of America, 4 Arkansas RAPPS, LLC, Little Rock, Arkansas, United States of America, 5 ARCare, Little Rock, Arkansas, United States of America, 6 Department of Health Services, Policy & Practice, Brown University, Providence, Rhode Island, United States of America, 7 Department of Behavioral and Social Sciences, Brown University School of Public Health, Providence, Rhode Island, United States of America

* jun_tao@brown.edu

## Abstract

Access to HIV care is a persistent challenge in rural settings. Arkansas has one of the highest proportions of rural residents in the U.S., yet data on statewide engagement and retention in HIV care are limited. This study used the Arkansas All-Payer Claims Database (APCD), a longitudinal database of medical, pharmacy, and enrollment claims from Medicaid and most commercial insurers, to evaluate the HIV care continuum and spatial access to providers. We identified people living with HIV (PLWH) from 2013–2017 using an algorithm requiring ≥2 HIV diagnosis claims or ≥1 prescription for antiretroviral therapy (ART). Engagement in care was defined as ≥1 CD4 or viral load (VL) test in a calendar year, while retention was defined as ≥2 tests in a year. Geospatial accessibility to HIV care at the county level was assessed using ArcGIS. A total of 5,033 PLWH in Arkansas were identified across the 5-year period. Only 18–21% of this population received CD4 testing and 47–51% received VL testing annually. ART prescription fills ranged from 61–88% annually. HIV testing rates among the broader insured population were also low, particularly in rural counties. There were 92 unique physicians with HIV-related care claims during the study period, equating to approximately 1 provider per 50 PLWH. Spatial access and care engagement were lower in rural counties. Significant gaps exist in HIV care engagement and retention in Arkansas, with notable gender and rural/urban disparities. Targeted interventions are crucial to address these disparities and end the HIV epidemic in the state.

**Data availability statement:** The data used in this analysis is commercial claims data, of which the authors do not have authorization to share. Interested parties must purchase or obtain this data through the Arkansas All-Payer Claims Database, managed by the Arkansas Center for Health Improvement (501) 526-2244). The authors did not receive any special privileges in accessing the data that other researchers would not have.

**Funding:** The study was supported by National Institute of Health (NIH) grants K01MH19660, R25MH083620, and the Providence/Boston Center for AIDS Research (P30AI042853).

**Competing interests:** The authors have declared that no competing interests exist.

## Introduction

While HIV incidence has stabilized in the United States overall, it remains high in the Southern U.S. [1]. Over half of new HIV diagnoses in 2021 were concentrated in the South [1]. Of the 1.2 million people living with HIV/AIDS (PLWH) in the US, more than 50% live in the South, despite Southern residents comprising only 38% of the total U.S. population [1]. The CDC defines the South as including the states of: Alabama, Arkansas, Delaware, Florida, Georgia, Kentucky, Louisiana, Maryland, Mississippi, North Carolina, Oklahoma, South Carolina, Tennessee, Texas, Virginia and West Virginia [1]. Like many Southern states, Arkansas is largely rural (99% of its land area is considered rural) and has high rates of poverty (15.4% in 2017–substantially higher than the national average of 12.3%) [2]. In 2017, over 40% of Arkansas residents, and nearly 50% of the 6,087 PLWH in Arkansas, lived in rural areas of the state [2,3]. By comparison, only about 20% of the U.S. population is classified as rural according to the 2020 Census, highlighting Arkansas' disproportionately rural and potentially underserved population [4]. Given its substantial rural HIV/AIDS burden, Arkansas is one of the seven states prioritized in the federal government's initiative "Ending the HIV Epidemic: A Plan for America" (EtHE) [5].

The HIV care continuum provides a framework for tracking progress in HIV care, including five main stages: diagnosis, linkage to care, retention in care, adherence to antiretroviral therapy (ART), and viral suppression [6]. Significant gaps still exist across the HIV care continuum in Arkansas. This includes gaps in HIV testing, which is essential for early diagnosis and linkage to care. In Arkansas, HIV testing is mandated in certain contexts, including prenatal care and upon entry to correctional facilities, in alignment with CDC recommendations [7]. However, HIV testing rates are suboptimal in Arkansas; in a 2017 national survey, only 34% of Arkansas residents reported ever having an HIV test, compared to the national average of 46% [8,9]. In urban areas, 35.8% of residents report testing, compared with only 30.9% in rural areas [10]. Limited access to care and HIV stigma contributes to suboptimal HIV testing rates in Arkansas [11]. An estimated 15% of PLWH in Arkansas are unaware of their infection [11]. Nearly half of PLWH in Arkansas rapidly progressed to AIDS in 2017, considerably higher than the national average of approximately 23% during the same period [12], suggesting that many individuals are diagnosed late in the course of their infection [3].

Another critical shortfall in Arkansas's HIV care continuum is engaging PLWH in HIV care. For PLWH to reach an undetectable viral load (rendering them unable to sexually transmit the virus), they must actively participate in HIV care. This involves steps such as initiating care, consistently attending appointments, undergoing antiretroviral therapy (ART), and maintaining ongoing involvement in their treatment plan. Over 80% of PLWH in Arkansas are not retained in care – based on the Health Resources and Service Administration (HRSA)/Ryan White definition of attending two HIV care visits within 12 months – with Black/African American (B/AA) individuals accounting for over 40% of this group [13]. Understanding factors influencing these disparities in outcomes is important for developing effective interventions to enhance retention in HIV care.

Importantly, limitations in statewide HIV data integration complicate a full characterization of the HIV care continuum, including HIV testing and engagement in HIV care. Although approximately one-third of the estimated 6,087 PLWH in the state are served by the Ryan White HIV/AIDS Program (RWHAP), a federally funded initiative in the US to provide comprehensive care and supportive services to uninsured and underinsured PLWH [14], clinical and laboratory data on these individuals are maintained in separate reporting systems (e.g., CAREWare) and are not routinely integrated into public surveillance reports or claims databases. Although Arkansas mandates the reporting of CD4 and viral load results to public health authorities, these data are submitted through surveillance systems such as eHARS and may not be linked to broader datasets. As a result, key information on HIV care and treatment, especially for uninsured individuals, may be fragmented or missing from statewide assessments. A more comprehensive and unified data approach is needed to accurately identify gaps and guide interventions across the care continuum.

One potential source of data to improve characterization of the HIV care continuum in Arkansas is the state's All-Payer Claims Database (APCD). APCDs are different in every state, but they are often comprehensive databases that include insurance claims on residents with Medicaid, fully insured commercial coverage, and some self-insured commercial coverage. Data from uninsured people receiving care through Ryan White programs, which is not an insurance program, are not included. Arkansas implemented an APCD in 2013. To our knowledge, APCDs have not been previously used to evaluate the HIV care continuum. We used the Arkansas APCD to characterize the HIV care continuum, identify gaps across the care continuum, and explore factors that influence engagement and retention in HIV care.

## Materials and methods

### Data sources

We used the Arkansas All-Payer Claims Database (APCD) to characterize HIV testing, and HIV-care engagement and retention from 2013 to 2017. We accessed the data between August 27th, 2019 to December 10th, 2024. The Arkansas APCD is a statewide repository of medical and pharmacy claims maintained by the Arkansas Center for Health Improvement (ACHI). An estimated 90% of Arkansas residents were covered by some form of insurance following Medicaid expansion in 2013, suggesting relatively high population-level coverage in the APCD [15]. As with other APCDs across the US, the Arkansas APCD includes information on enrollment, medical claims, pharmacy claims, and provider data from private insurers and Medicaid, and is updated monthly. Only services billed to insurance are captured; laboratory results and clinical notes are not included. Although the Arkansas APCD included Medicare data during 2013–2017, these claims were not fully standardized or integrated into the main APCD structure. As a result, Medicare data were incomplete and not usable for reliable analysis of HIV care outcomes. Individuals aged 65 and older may appear in the dataset, but their service utilization data (e.g., labs, ART prescriptions) may be missing or underreported. Additionally, we are not able to capture PLWH who received free HIV care and/or ART medicine through the AIDS Drug Assistance Program, offered by the Ryan White HIV/AIDS Program.

This study used de-identified data from the Arkansas All-Payer Claims Database and was reviewed by the University of Arkansas for Medical Sciences (UAMS) Institutional Review Board. The study was determined to be exempt from IRB oversight on August 27, 2019 (UAMS IRB #239879). Need for consent was waived by the ethics committee. The analysis and manuscript were reviewed and approved by the Arkansas Center for Health Improvement (ACHI), which oversees the AR APCD. All analyses were conducted in compliance with relevant ethical guidelines and data use agreements. Data access was restricted to authorized personnel and performed on secure servers using password-protected devices with authentication protocols.

### Study population

We identified around 1.3 million members with full coverage in each calendar year (60 consecutive months) in the Arkansas APCD during 2013–2017. We included individuals with full medical and pharmacy coverage for the entire

duration of any given calendar year during 2013–2017. This allowed individuals to contribute data for one or more years depending on their coverage. HIV testing, service utilization, and care engagement outcomes were assessed within the calendar year(s) for which full coverage was available. We then summarized these yearly outcomes to report overall patterns across the study period. We also identified all HIV testing claims and individuals who were tested in each calendar year. Individuals with a diagnosis of HIV/AIDS were identified as having at least two claims containing an ICD-9/10 code or a healthcare common procedure code (HCPCS) for HIV/AIDS over 30 days apart and/or one or more pharmacy claims for ART medication [16]. This HIV identification algorithm is commonly used in administrative claims analyses and has been employed in prior studies using MarketScan and Medicaid data [17,18]. This subgroup of PLWH was used to assess engagement and retention in the HIV care continuum in Arkansas. Because medications commonly used to treat HIV are also used to treat hepatitis B, for patients who only ever picked up antiretroviral therapy medication, we further excluded those with a diagnosis of hepatitis B. We excluded NDC codes associated with medications used exclusively for pre-exposure prophylaxis (PrEP), such as Truvada and Descovy. Only antiretroviral medications indicated for HIV treatment were included in ART identification, ensuring that individuals on PrEP were not misclassified as PLWH.

## Measurement of engagement and retention in the HIV care continuum

We used an algorithm to define the cohort of individuals living with HIV/AIDS, based on HIV-related diagnosis codes, ART prescriptions, and laboratory tests (CD4 or viral load) (Fig 1). The APCD does not contain laboratory test results (e.g., CD4 count or viral load values), nor clinical notes or provider-reported data. As a result, we inferred HIV care engagement using claims for relevant procedures and services (e.g., HIV testing, CD4 test billing codes), rather than direct results or outcomes. The ICD-9/10 and HCPCS codes for the HIV care services are presented in S1 Table. As per CDC criteria, engagement in HIV care was defined as at least one CD4 or viral load test within the same calendar year [19]. Retention in HIV care was defined as having two or more CD4 or viral load tests within the same calendar year [19]. Although we did not impose a minimum time interval between tests, each claim was captured as a unique service line, which minimizes the risk of double-counting multiple tests from a single clinical encounter.

## Identification of physicians who provided HIV/AIDS care

We identified HIV/AIDS care providers located in Arkansas using two approaches. First, we identified Ryan White contracted HIV/AIDS providers (e.g., physicians, nurse practitioners, and physician assistants) using data from the Arkansas Department of Health. Second, using the Arkansas APCD, we identified all physicians who provided at least one of the following HIV-related services: prescribing ART medication, ordering a CD4 count test, or ordering an HIV viral load test. We extracted office addresses, Type-1 national provider identifiers (individual practice), and Type-2 national provider identifiers (group practice). Duplicate providers were removed, and the remainder of the Type-1 and Type-2 identifiers were merged with office addresses. Seventy-two physician providers were identified using the first approach, and 50 physician providers using the second approach.

## Statistical analysis

We identified antiretroviral drug prescriptions from pharmacy claims, as well as CD4 tests, viral load tests, and general clinical visits from medical claims (S1 Table). A unique identifier was used to link each individual's claims in the dataset. We used the average age across the study period to offset the shifting of age group across years. Using descriptive statistics, we show rates at which residents present for HIV testing and engagement and retention by gender, age, rural/urban, and calendar year in the HIV care continuum at the county level. Rural and urban classification was based on the U.S. Census Bureau's 2010 urban–rural definition at the county level [20]. Counties were grouped into three categories:

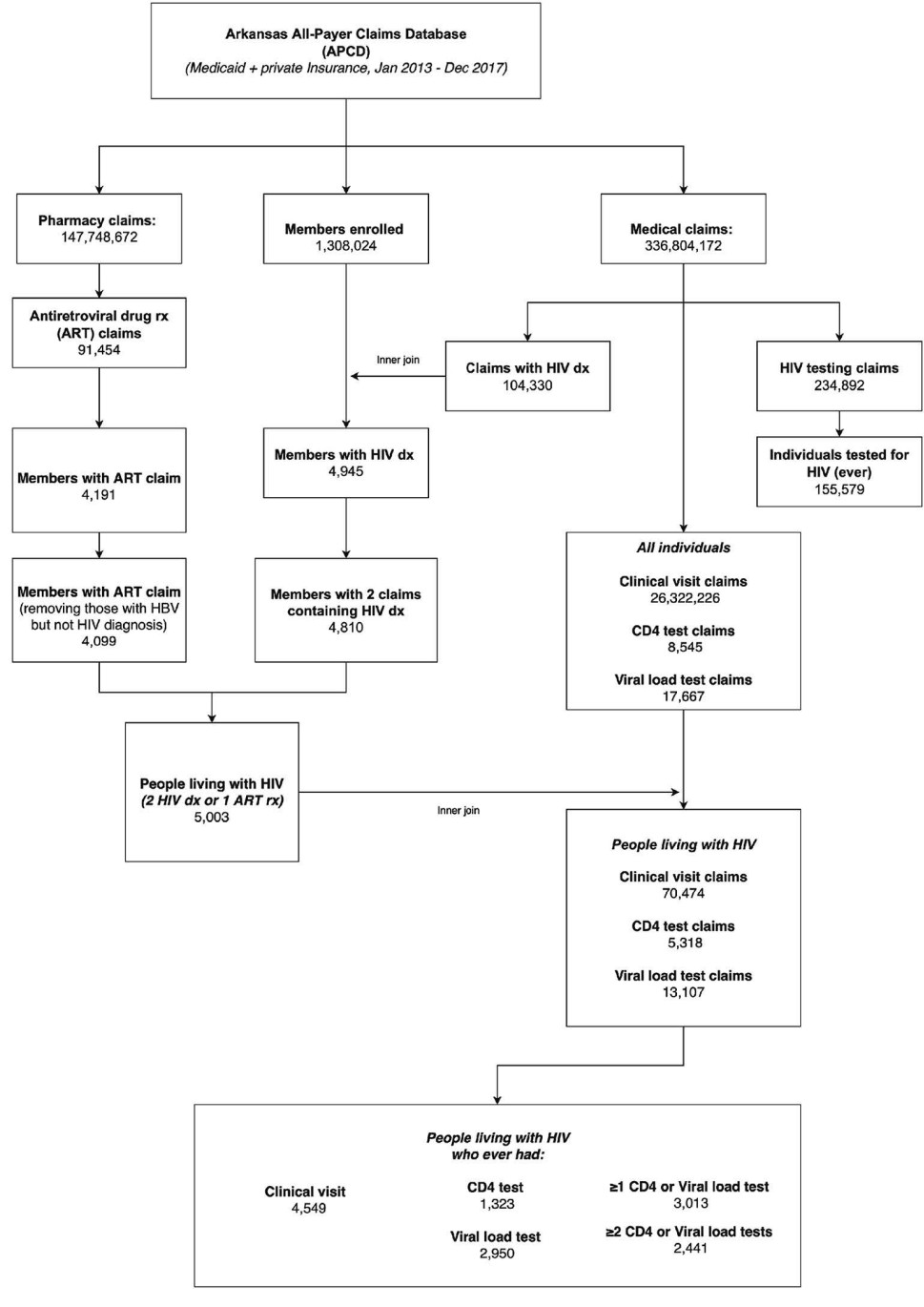

**Fig 1. Statistical analysis flowchart.**

completely rural (100% of the population living in rural areas), mostly rural (50.0–99.9% rural), and mostly urban (less than 50.0% rural). We used Chi-square tests to see if there were any meaningful differences in the distribution of HIV care outcomes and certain categorical variables. Additionally, we conducted a Cochran-Armitage Trend Test to find out if there was a consistent increase or decrease in the rate of HIV care outcomes over the time of the study.

## Spatial accessibility analysis

We summarized county-level HIV care engagement outcomes using individual-level claims data from 2013 to 2017. Each claim was linked to the member file using the ZIP code of residence for that year, allowing for time-updated residence assignment. For spatial analysis, we aggregated these outcomes across all five years to generate cumulative counts and rates at the county level. We then merged this county-level HIV care engagement data with the Arkansas county-level shapefile. ArcGIS 10.6 (ESRI 2018. ArcGIS Desktop: Release 10.6. Redlands, CA: Environmental Systems Research Institute) was used to generate a map of this engagement data. We identified clinics and providers who provided care for PLWH. The two-step floating catchment area (2SFCA) method was used to assess spatial HIV/AIDS healthcare accessibility [21]. First, the provider-to-patient ratio was estimated based on the number of providers assigned to the county centroid – the center point of a specific area – and the number of PLWH within that centroid's one and a half hours drive time catchment. The provider-to-patient ratio was assigned to the entire catchment area. Second, we calculated the supply ratio for a specific location in need by adding together the provider-to-patient ratios from all nearby provider areas that cover this location. This approach helps us understand how well the location is served by healthcare providers based on the number of providers available compared to the number of patients in the area. The interpretation of these catchment settings is that individuals with an HIV/AIDS diagnosis in a certain county have access to HIV care providers that are within a buffer of predefined traveling distance. The supply ratios, which measure the availability of healthcare services, were calculated for the entire high-need area. This area is identified by its significant lack of healthcare services, where the number of patients needing care greatly exceeds the number of available healthcare providers. The final accessibility score is a composite that accounts for the capacity of providers, the population of the demanding group, and the distance between them. ArcGIS was used to map the supply ratios at the county level.

## Results

### HIV testing

We identified the number of individuals who were tested for HIV in each calendar year (S1 Table). Only 11.7% of individuals in the Arkansas APCD dataset were tested for HIV during 2013–2017. Among individuals who were tested for HIV, the majority were female (79%), aged 15–34 (69%), and lived in a mostly urban area (65%).

### Engagement and retention in HIV care

This study identified 83% of PLWH (5,033 out of 6,087) in Arkansas using the APCD data during 2013–2017. The majority of identified PLWH in the dataset were male (68%), in the age group of 25–54 (around 66% across 2013–2017), lived in areas that are mostly urban (74%), and had private insurance (46%) (S1 Table). Among PLWH in Arkansas identified in the Arkansas APCD between 2013 and 2017 based on ≥2 HIV diagnosis claims or ≥1 ART prescription fill within a calendar year, an average of 50.9% were engaged in care, and an average of 45.2% were retained in care. Women were less likely than men to engage with and be retained in HIV care ($p < 0.05$). PLWH living in mostly urban areas had better engagement and retention in the HIV care continuum than those living in rural areas, with urban PLWH being 56% engaged with and 35% retained in care, compared to completely rural PLWH being 51% engaged and 31% retained. Children with HIV/AIDS (less than 15 years old) and older people (over 54 years old) were less likely to engage and be retained in HIV care, compared to adults aged 24–35. Trends in the HIV care continuum are presented in Fig 2.

### Health care use

Among 5,033 PLWH living in Arkansas during 2013–2017, annual estimates showed that 68% to 73% had at least one clinical visit, 61% to 88% picked up at least one prescription for ART medication, 18% to 21% had at least one CD4 count

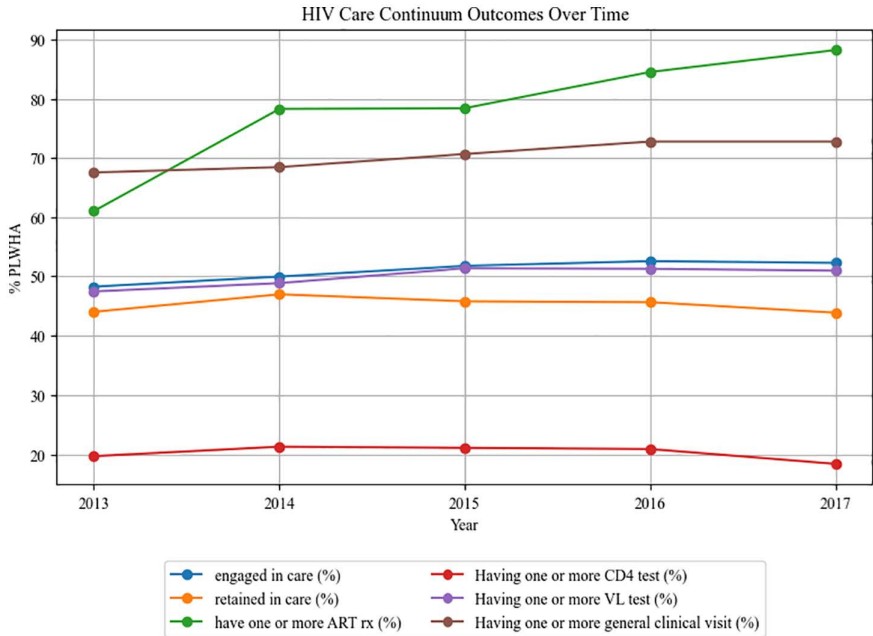

**Fig 2. HIV care continuum outcomes over time.** Trends in the HIV care continuum between 2013 and 2017 in Arkansas.

test, and 47% to 51% had at least one viral load test (S1 Table and Fig 2). Women were significantly less likely to have picked up a prescription for ART medication or receive CD4 and/or viral load testing than men (all $p < 0.001$). PLWH in completely rural areas were less likely to have picked up a prescription for ART medication and/or have a CD4 test than PLWH in mostly urban areas ($p = 0.007$ and $0.04$, respectively). PLWH in the youngest and oldest age groups had lower HIV care utilization than other age groups. A slight increase was observed in the usage of HIV health care in Arkansas from 2013 to 2017, including engagement in HIV care ($p = 0.035$), ART prescription pickup ($p < 0.001$), viral load testing ($p = 0.004$), and general clinical visits ($p < 0.001$).

### The spatial distribution of HIV cases and health providers

We identified 99 HIV care providers in Arkansas, including one physician assistant, six nurse practitioners, and 92 physicians. The HIV provider-to-patient ratio was 50–1. Fig 3 shows the spatial distribution of HIV cases and HIV healthcare providers in Arkansas. Three HIV case clusters can be observed from the map, centering around Pulaski, Washington, and Crittenden counties. Pulaski County is home to Little Rock, the capital of Arkansas, Washington County encompasses Fayetteville, Arkansas's second most highly populated city, and Crittenden County is part of the Memphis metropolitan area. Most HIV cases are found in Pulaski County with a total of 1,804 cases. The rest of the counties have HIV case numbers under 300.

The spatial pattern of HIV healthcare providers corresponds to the findings of HIV cases, with 76% of HIV care providers located within the three identified areas with HIV cases. These areas of increased HIV cases centered around the three counties of interest, as indicated by the circles on the map (Fig 3). Areas on the map shaded in dark orange have the highest number of HIV cases, while areas shaded in pale orange have the lowest. Note that the size of the yellow marker indicating provider(s) locations corresponds to provider capacity, as a point may represent more than one provider. Most of the HIV providers were in urban areas (63%), with the majority located in Pulaski County.

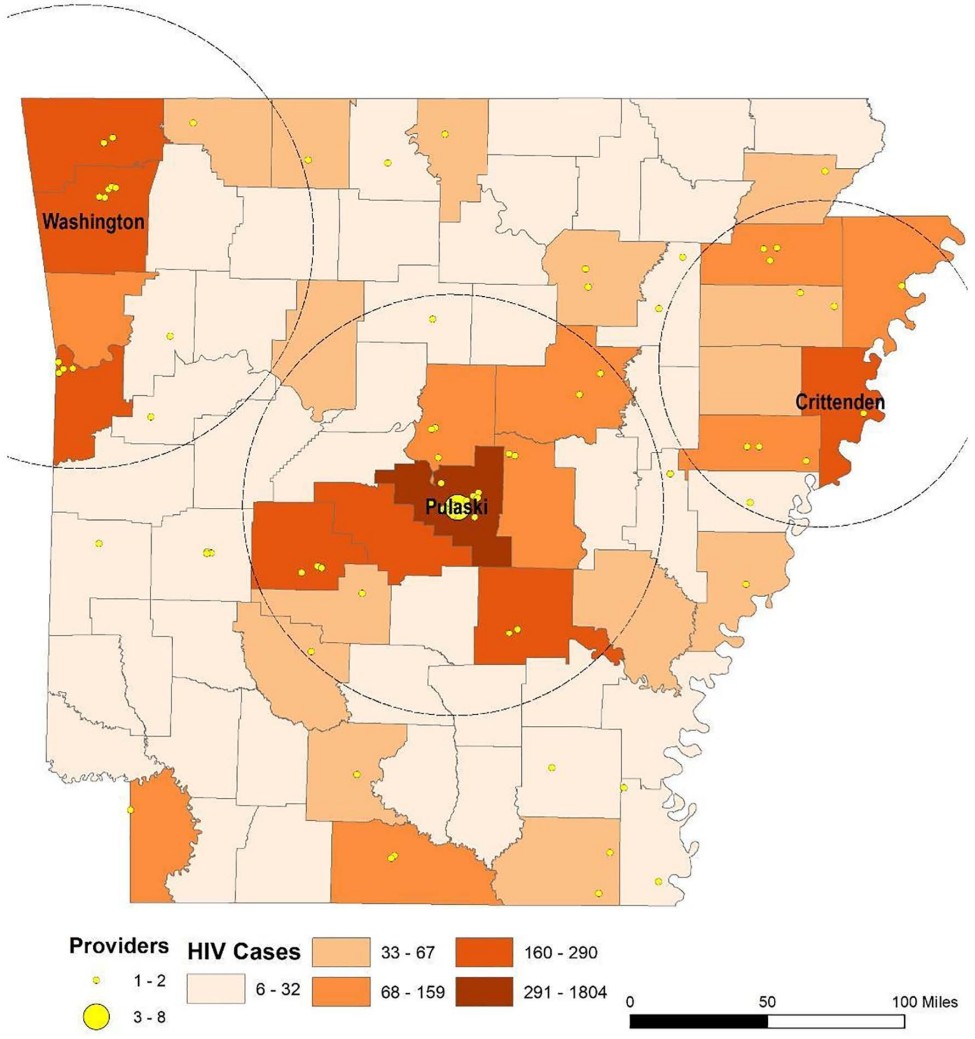

**Fig 3. The spatial distribution of HIV cases and health providers.** Maps were created in ArcGIS Pro (Esri, Redlands, CA). Base-map: Esri, HERE, Garmin, © OpenStreetMap contributors, and the GIS User Community. https://www.arcgis.com/home/item.html?id=e4191ebc440a4789b3016c54f8651e54.

## Spatial accessibility analysis

As introduced in the methods section, a two-step floating catchment area (2SFCA) method was applied to the data to measure accessibility at differing catchment sizes. Four catchment sizes were used in this study, including 15, 30, 60, and 120 miles. Fig 4 shows the county-level spatial accessibility maps generated from these four catchment sizes. The counties shaded with a darker blue indicate better access to HIV health resources than counties with a lighter blue. Health resources are only available to a county itself and/or its immediate neighboring counties.

A catchment of 15 miles in 2SFCA, shown in Fig 4(a), resulted in most counties (56%, or 42 out of 75) having an accessibility score of zero, which means there is no access to health providers at all due to the limited travel distance. As the catchment size increases (from 15 miles to 30, 60, and 120 miles), the "access desert" areas, where the accessibility

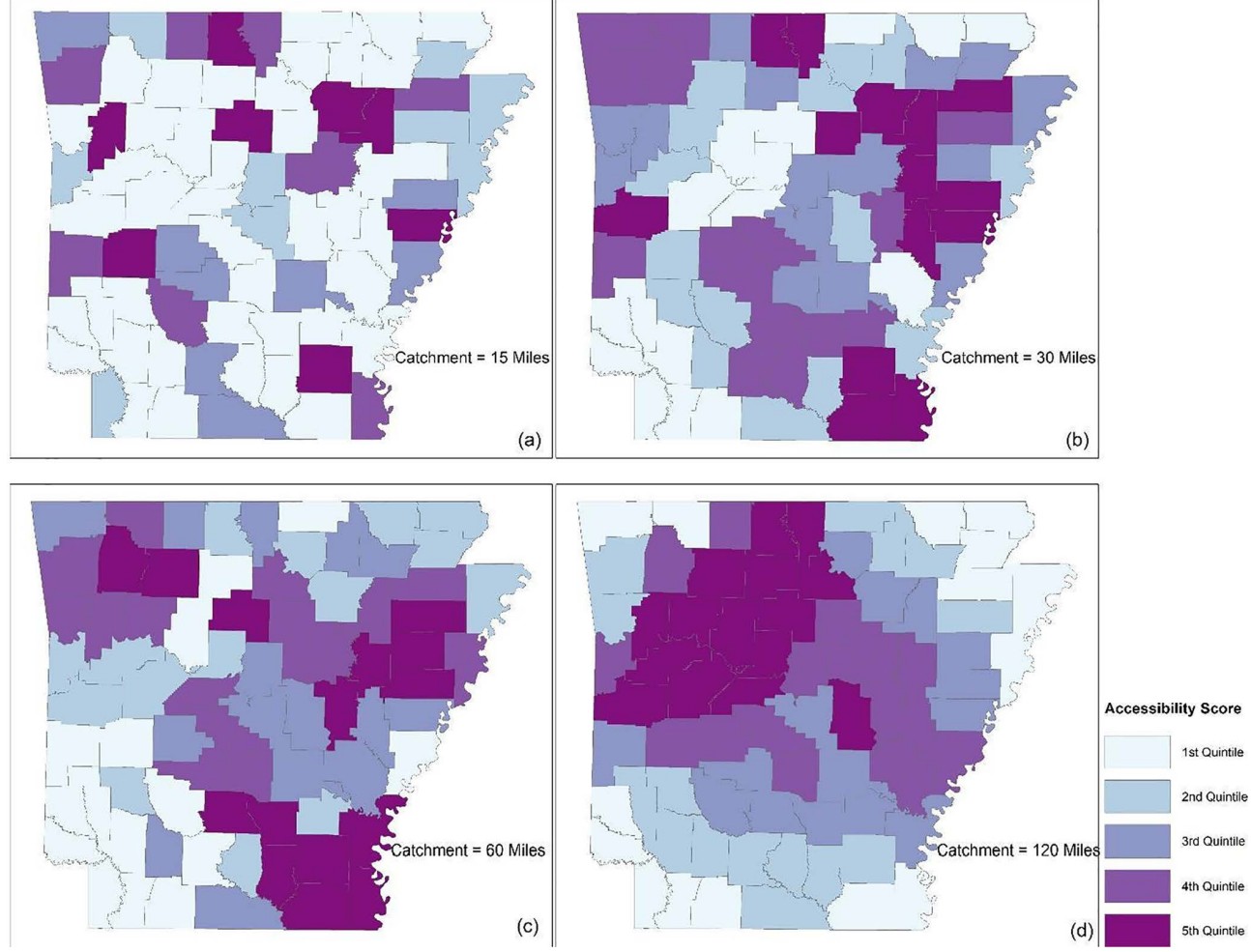

**Fig 4. Spatial accessibility map at a county level.** Maps were created in ArcGIS Pro (Esri, Redlands, CA). Basemap: Esri, HERE, Garmin, © Open-StreetMap contributors, and the GIS User Community. https://www.arcgis.com/home/item.html?id=e4191ebc440a4789b3016c54f8651e54.

score equals zero, shrink significantly. There are only nine counties left with zero access when the catchment size is 30 miles, including Yell, Sevier, Howard, Hempstead, Columbia, Arkansas, Pope, Randolph, and Fulton counties. All counties have access when the catchment size is 60 miles. Areas with better access to health resources are located in the northwest, east and southeast regions of Arkansas.

A catchment of 120 miles results in a completely different pattern. Fig 4(d) shows that the "access desert" areas identified in catchment settings of 15, 30, and 60 miles become the best areas in terms of healthcare access. As the catchment size increases to 120 miles or even larger, almost every county achieves access to all the health providers in Arkansas.

## Discussion

This is one of the first studies to comprehensively assess the HIV care continuum in a rural southern state using APCD data. APCDs are comprehensive insurance claim databases that have the potential to provide important information on HIV care continuum outcomes and supplement traditional approaches like public health HIV surveillance and Ryan White program data. A major gap in public health surveillance is that frequently, only HIV-positive tests are reported. Therefore,

the number of HIV tests being performed is often unclear. In this analysis of a rural Southern state in the US, only 12% of insured members had a documented HIV test over the study period based on billing claims, and people who lived in urban areas were more likely to have an HIV test than those living in rural areas. While these results may underestimate testing that occurs outside of billable clinical encounters (e.g., through health departments, community outreach, or home tests), they are directionally consistent with prior survey-based estimates. The 2017 Behavioral Risk Factor Surveillance System (BRFSS) reported that 34% of Arkansas adults had ever received an HIV test, with testing less common in rural areas [8–10]. These survey-based estimates reflect lifetime testing and self-report, while our data capture testing within insured clinical settings during enrollment. Taken together, these findings suggest that a substantial proportion of Arkansans—especially in rural areas—may remain untested for HIV. Effective strategies to improve testing uptake are needed, particularly in medically underserved communities. Limited access to HIV prevention and care services may be driving lower testing rates in rural areas. People in rural areas may also have limited HIV knowledge and low self-perceived HIV risk [22]. As the HIV testing rate remains low in Arkansas, especially in rural areas, effective interventions are needed to promote HIV testing in rural Arkansas [10]. Future approaches should consider using APCDs to inform public health interventions focusing on ending the HIV epidemic in the US.

Engagement and retention in HIV care are critical to achieve viral suppression at the individual level and to reduce HIV secondary transmission at the population level [21,23]. However, little is known about the HIV care continuum in Arkansas, as no statewide monitoring of PLWH receiving HIV care exists. Ryan White could provide engagement in HIV care only among a covered minority subset of the total PLWH in the state. In this study, APCDs were utilized to fill this knowledge gap. We used a big-data approach to conduct a systematic and comprehensive evaluation of engagement in the HIV care continuum. Significant gaps were observed in the HIV care continuum in Arkansas. From 2013 to 2017, an average of 51% of PLWH were engaged in HIV care, and an average of 45% of PLWH were retained in HIV care in Arkansas. Meanwhile, 74% of PLWH are engaged in HIV care, and 58% are retained in HIV care across the United States [19]. Therefore, PLWH in Arkansas had poorer outcomes for both engagement and retention in HIV care when compared to the national average.

Gender disparities existed in HIV testing, engagement and retention in HIV care, and the usage of HIV care. Although women are more likely to use other health care resources than men [24–26], a similar pattern was not observed in terms of HIV care. Women were more likely to get an HIV test, mainly because HIV testing during pregnancy is required by law in Arkansas [27]. However, women were significantly less likely to engage and retain in HIV care and use HIV care health resources in this study. This disparity has also been reported by other studies conducted in the US [18,28]. Although women only account for 16% of new HIV infections in 2018 [11], tailored interventions should be implemented to improve engagement and retention in HIV care for women.

In addition, a geospatial accessibility analysis was performed to present the geospatial pattern of engagement in HIV care. Of 5,033 PLWH, 14% lived in rural communities. These rural PLWH were more likely to have poorer outcomes of engagement in HIV care than their urban counterparts. Most HIV/AIDS providers were concentrated in Pulaski, Washington, and Crittenden counties. These three areas are all considered urban settings. Rural/urban health disparities in accessing HIV/AIDS care have also been reported in other Southern states [29]. However, driving long distances to HIV/AIDS providers likely presents major barriers for engagement and retention in HIV care. Several interventions could potentially address disparities in the HIV care continuum by demographic and geographic subgroups. For example, expansion of telemedicine services, mobile health (mHealth) tools, and peer navigation models have shown effectiveness in promoting linkage and retention. Tailored outreach strategies targeting men, racial/ethnic minorities, and individuals with low perceived risk may also help increase HIV testing and earlier diagnosis. Integrating such interventions into state-level efforts may help reduce the urban-rural gap in HIV outcomes observed in this study.

Our study has limitations. First, claims data were used as a proxy to estimate engagement in HIV care in this study. Because the capturing of HIV care using APCD depends on billing and claims, we were not able to capture HIV services

provided without billing. For example, individuals could get a free HIV test from the Arkansas Department of Health, clinics, and community-based organizations and purchase home-based HIV testing kits in most pharmacy stores without billing their insurances. These alternative test pathways likely contributed to the lower testing rate in this study. Similarly, we were not able to identify uninsured PLWH who may have had different experiences in HIV/AIDS care. While Arkansas has implemented Medicaid expansion since 2013 and over 90% of residents were insured, approximately one third of PLWH may be omitted from this analysis but could receive HIV/AIDS care through the Ryan White program, which is not captured in APCD. The exclusion of this population likely contributed to underestimation of HIV care engagement and service utilization in our analysis. Additionally, although Medicare data were technically available in the Arkansas APCD during the study period, they were not transformed into the unified APCD format and were often incomplete. This limitation likely led to underrepresentation of HIV care outcomes among Medicare beneficiaries, especially those aged 65 and older. People experiencing coverage disruptions—such as those affected by Medicaid churn—may also have been underrepresented, potentially biasing the results toward individuals with more stable insurance coverage. Furthermore, our analysis only included providers located in Arkansas and may not fully capture care received across state lines. This is particularly relevant for eastern Arkansas residents who may access HIV care in Memphis, Tennessee—a major regional care hub. As a result, our estimates of care engagement may underestimate service use in border regions. Lastly, due to limited geographic information in the Arkansas APCD, we were unable to assess the effect of individual-level factors, such as race and ethnicity, on engagement and retention in HIV care.

## Conclusions

These study findings provide important information to address the HIV epidemic and allocate HIV care resources in a Southern state with high rates of HIV infection. Our study is the first comprehensive evaluation of characterizing the HIV care continuum in Arkansas. HIV testing, engagement and retention in HIV care, and the usage of HIV care services in Arkansas were lower than the national average. Arkansas has a significant HIV burden among rural areas. Rural PLWH had worse outcomes of engagement in HIV care. Women living with HIV tended to have worse outcomes of HIV care than their male counterparts. Several intervention strategies could be feasible to achieve improvement in outcomes in Arkansas, including promoting HIV testing, training primary care providers to provide HIV/AIDS health care services, allocating HIV/AIDS health care resources to communities that bear significant HIV burden, assisting PLWH in linking to care and helping them retain in care, and/or expanding telemedicine services in existing HIV/AIDS clinics. Innovative implementation strategies are needed to expand HIV/AIDS care to PLWH who need it most.

## Supporting information

**S1 Table. HIV testing and engagement in the HIV care continuum in Arkansas, 2013–2017.**
(DOCX)

## Author contributions

**Conceptualization:** Jun Tao, Mofan Gu, S. Alexandra Marshall, Jure Baloh, Mike Cima, Cornelius Mabin, Nickolas D. Zaller, Dan Moore.

**Formal analysis:** Jun Tao, Mofan Gu.

**Funding acquisition:** Jun Tao.

**Methodology:** Jun Tao, Mofan Gu, S. Alexandra Marshall, Jure Baloh, Mike Cima, Cornelius Mabin, Nickolas D. Zaller, Dan Moore.

**Project administration:** Jun Tao.

**Supervision:** Jun Tao.

**Writing – original draft:** Jun Tao.

**Writing – review & editing:** Jun Tao, Mofan Gu, S. Alexandra Marshall, Jure Baloh, Mike Cima, Cornelius Mabin, Nickolas D. Zaller, Dan Moore, Hannah Parent, Heather Horton, Ira B. Wilson, Amy S. Nunn, Philip A. Chan.

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
