## [Decision Letter · Decision Letter 0]

1 Oct 2025

Dear Dr. Tao,

Thank you for submitting your manuscript to PLOS ONE. After careful consideration, we feel that it has merit but does not fully meet PLOS ONE’s publication criteria as it currently stands. Therefore, we invite you to submit a revised version of the manuscript that addresses the points raised during the review process.

We look forward to receiving your revised manuscript.

Kind regards,

Lakshminarayana Chekuri, MD, PhD

Academic Editor

PLOS ONE

 [The study was supported by National Institute of Health (NIH) grants K01MH19660, R25MH083620, and the Providence/Boston Center for AIDS Research (P30AI042853).]. 

3. For studies involving third-party data, we encourage authors to share any data specific to their analyses that they can legally distribute. PLOS recognizes, however, that authors may be using third-party data they do not have the rights to share. When third-party data cannot be publicly shared, authors must provide all information necessary for interested researchers to apply to gain access to the data. (https://journals.plos.org/plosone/s/data-availability#loc-acceptable-data-access-restrictions)

4. We note that Figures 3 and 4 in your submission contain [map/satellite] images which may be copyrighted. All PLOS content is published under the Creative Commons Attribution License (CC BY 4.0), which means that the manuscript, images, and Supporting Information files will be freely available online, and any third party is permitted to access, download, copy, distribute, and use these materials in any way, even commercially, with proper attribution. For these reasons, we cannot publish previously copyrighted maps or satellite images created using proprietary data, such as Google software (Google Maps, Street View, and Earth). For more information, see our copyright guidelines: http://journals.plos.org/plosone/s/licenses-and-copyright.

1. You may seek permission from the original copyright holder of Figures 3 and 4 to publish the content specifically under the CC BY 4.0 license. 

Additional Editor Comments:

Thank you for your scholarly contribution. I'd also like to thank the authors for choosing PLOS ONE to publish your findings from this study. Comments from reviewers are provided below. Please review these comments and I suggest address them and resubmit your manuscript. Your timely response would help this study be published and will make it accessible to interested readers across the world. I look forward to reviewing your revised manuscript. I wish you good luck with your future endeavors.

Reviewers' comments:

Reviewer's Responses to Questions

**Comments to the Author**

1. Is the manuscript technically sound, and do the data support the conclusions?

Reviewer #1: Yes

Reviewer #2: Yes

2. Has the statistical analysis been performed appropriately and rigorously?

Reviewer #1: Yes

Reviewer #2: I Don't Know

3. Have the authors made all data underlying the findings in their manuscript fully available?

Reviewer #1: Yes

Reviewer #2: Yes

4. Is the manuscript presented in an intelligible fashion and written in standard English?

Reviewer #1: Yes

Reviewer #2: Yes

Reviewer #1: Overall, this manuscript provided important insights into engagement and retention in HIV care in Arkansas using All Payer Claims Databases, which represents an alternative approach to assessing engagement and retention in the developed world. The limitations are appropriately acknowledged, and the results are presented cautiously without being overstated. The manuscript is also well structured and aligns well with research and publication ethics.

That said, I have a few observations that may help strengthen the study.

Focus of findings: While the study set out to assess engagement and retention in care, a significant portion of the findings and discussion focusses on HIV testing. Since HIV testing would typically precede engagement and retention in care, further clarification on how these results contribute to the primary outcomes of engagement and retention would be helpful.

Results section: under statistical methods, the manuscript mentions chi-square tests were used to examine meaningful differences; however, the supplementary table doesn't present the Chi-square results. Could the authors clarify whether these results are missing, presented elsewhere or were not statistically significant?

in summary, this is a well prepared manuscript that addresses an important topic in HIV care. These above few observations will improve the clarity and strength of the results.

Reviewer #2: This is a well written and valuable manuscript analyzing a data from the Arkansas All-payer Claims Database to characterize the HIV care cascade. Authors carefully document assumptions and criteria for extracting and coding cases. The study is important and worthy of publication. Authors are to be commended for developing a novel approach for exploring the HIV care continuum in Arkansas.

There are a number of minor issues that need to be addressed:

Throughout

PLoS One is a global publication, and ‘rural South’ needs to be explicitly unpacked both in the title and clearly in the introduction to the U.S. rural South.

Introduction

Please include supporting information such as the proportion of U.S. population in ‘the South’ and how you’re defining ‘the South’. Statistics about what proportion of the U.S. population lives in the south are needed to explain why it's problematic that >50% of PLWH in the U.S. live in the south.

For a global readership, when statements such as “high rates of poverty (15.4% in 2017)” line 50 are made, it would be helpful to give the U.S. median or comparison given that many countries have far higher rates of poverty than 15.4%.

Line 65: ‘rapidly progressed’ - please define and provide a U.S. comparison for states where new HIV dx are associated with less rapid progression.

Line 68 – ‘they must actively participate in care’ – there are PLWH who are virally suppressed and out of care; critical determinant is generally taking ART meds, not the entire chain of engagement you describe. Recommend rewording sentence to be more accurate.

Please provide a sentence introducing the Ryan White program for readers who may not be familiar with it.

Methods

Please define how you are using rural and urban since the definition varies across countries

Results

Line 192 – ‘ever tested for HIV’ – does this really mean ever tested, or does it mean tested during the four years captured by your data? Please be clear.

Line 197 – ‘in mostly urban’ – does this mean areas where people live are predominantly urban? Or that most people are living in urban areas. The rural/urban transition spaces of suburbs are different, and it’s not clear from your language what you mean.

Line 267 – ‘our analysis only reflects testing over the four year period covered in our dataset’

Limitations

You noted in lines 76-77 that approximately a third of PLWH in Arkansas receive care through the Ryan White program. A third is not ‘a small proportion’ (line 311). I recommend explicitly noting the approximately one third you’re potentially missing in the limitations (line 311), and also in the discussion (line 277-278) where you discuss the Ryan White program.

**Do you want your identity to be public for this peer review?** For information about this choice, including consent withdrawal, please see our Privacy Policy

Reviewer #1: No

Reviewer #2: **Yes: ** Debbie Humphries

---

## [Author Response · Author response to Decision Letter 1]

17 Oct 2025

Reviewer #1: Overall, this manuscript provided important insights into engagement and retention in HIV care in Arkansas using All Payer Claims Databases, which represents an alternative approach to assessing engagement and retention in the developed world. The limitations are appropriately acknowledged, and the results are presented cautiously without being overstated. The manuscript is also well structured and aligns well with research and publication ethics.

That said, I have a few observations that may help strengthen the study.

Focus of findings: While the study set out to assess engagement and retention in care, a significant portion of the findings and discussion focusses on HIV testing. Since HIV testing would typically precede engagement and retention in care, further clarification on how these results contribute to the primary outcomes of engagement and retention would be helpful.

Response: Thank you for this comment. In the manuscript, we highlight how HIV testing is crucial for early diagnosis and linkage to care. Diagnosis is the first stage in the HIV care continuum.

Results section: under statistical methods, the manuscript mentions chi-square tests were used to examine meaningful differences; however, the supplementary table doesn't present the Chi-square results. Could the authors clarify whether these results are missing, presented elsewhere or were not statistically significant?

in summary, this is a well prepared manuscript that addresses an important topic in HIV care. These above few observations will improve the clarity and strength of the results.

Response: Thank you for this comment. Supplement Table 1, as well as the Results section, have been revised to show results of chi-square tests and Cochran-Armitage trend test for HIV care utilization.

Reviewer #2: This is a well written and valuable manuscript analyzing a data from the Arkansas All-payer Claims Database to characterize the HIV care cascade. Authors carefully document assumptions and criteria for extracting and coding cases. The study is important and worthy of publication. Authors are to be commended for developing a novel approach for exploring the HIV care continuum in Arkansas.

There are a number of minor issues that need to be addressed:

Throughout

PLoS One is a global publication, and ‘rural South’ needs to be explicitly unpacked both in the title and clearly in the introduction to the U.S. rural South.

Response: Thank you for this comment. We have updated the title and introduction to include rural Southern U.S.

Introduction

Please include supporting information such as the proportion of U.S. population in ‘the South’ and how you’re defining ‘the South’. Statistics about what proportion of the U.S. population lives in the south are needed to explain why it's problematic that >50% of PLWH in the U.S. live in the south.

Response: Thank you for this comment. We have updated the introduction to include the suggested information.

For a global readership, when statements such as “high rates of poverty (15.4% in 2017)” line 50 are made, it would be helpful to give the U.S. median or comparison given that many countries have far higher rates of poverty than 15.4%.

Response: Thank you for this comment. We have updated the introduction to include this statistic.

Line 65: ‘rapidly progressed’ - please define and provide a U.S. comparison for states where new HIV dx are associated with less rapid progression.

Response: Thank you for this comment. We have updated the introduction to include the suggested information.

Line 68 – ‘they must actively participate in care’ – there are PLWH who are virally suppressed and out of care; critical determinant is generally taking ART meds, not the entire chain of engagement you describe. Recommend rewording sentence to be more accurate.

Response: Thank you for this comment. We have reworded this sentence as suggested.

Please provide a sentence introducing the Ryan White program for readers who may not be familiar with it.

Response: Thank you for this comment. We have updated the introduction to include a definition of RWHAP.

Methods

Please define how you are using rural and urban since the definition varies across countries

Response: Thank you for this comment. We have included the rural and urban definitions in the Methods section.

Results

Line 192 – ‘ever tested for HIV’ – does this really mean ever tested, or does it mean tested during the four years captured by your data? Please be clear.

Response: Thank you for this comment. We have clarified that we meant testing during the four years captured by data.

Line 197 – ‘in mostly urban’ – does this mean areas where people live are predominantly urban? Or that most people are living in urban areas. The rural/urban transition spaces of suburbs are different, and it’s not clear from your language what you mean.

Response: Thank you for this comment. We have revised this sentence to be more clear.

Line 267 – ‘our analysis only reflects testing over the four year period covered in our dataset’

Limitations

You noted in lines 76-77 that approximately a third of PLWH in Arkansas receive care through the Ryan White program. A third is not ‘a small proportion’ (line 311). I recommend explicitly noting the approximately one third you’re potentially missing in the limitations (line 311), and also in the discussion (line 277-278) where you discuss the Ryan White program.

Response: Thank you for this comment. We have made the suggested revisions.

---

## [Decision Letter · Decision Letter 1]

8 Dec 2025

Engagement and retention in HIV care in rural Southern U.S. using an All-Payer Claims Database

PONE-D-25-39217R1

Dear Dr. Tao,

We’re pleased to inform you that your manuscript has been judged scientifically suitable for publication and will be formally accepted for publication once it meets all outstanding technical requirements.

Kind regards,

Lakshminarayana Chekuri, MD, PhD

Academic Editor

PLOS One

Additional Editor Comments (optional):

Reviewers' comments:

Reviewer's Responses to Questions

**Comments to the Author**

Reviewer #1: All comments have been addressed

Reviewer #2: All comments have been addressed

2. Is the manuscript technically sound, and do the data support the conclusions?

Reviewer #1: (No Response)

Reviewer #2: Yes

3. Has the statistical analysis been performed appropriately and rigorously?

Reviewer #1: Yes

Reviewer #2: I Don't Know

4. Have the authors made all data underlying the findings in their manuscript fully available?

Reviewer #1: (No Response)

Reviewer #2: Yes

5. Is the manuscript presented in an intelligible fashion and written in standard English?

Reviewer #1: (No Response)

Reviewer #2: Yes

Reviewer #1: (No Response)

Reviewer #2: (No Response)

**Do you want your identity to be public for this peer review?** For information about this choice, including consent withdrawal, please see our Privacy Policy

Reviewer #1: **Yes: ** Victoria Babirye Tumusiime

Reviewer #2: **Yes: ** Debbie Humphries

---

## [Editor Report · Acceptance letter]

PONE-D-25-39217R1

PLOS One

Dear Dr. Tao,

I'm pleased to inform you that your manuscript has been deemed suitable for publication in PLOS One. Congratulations! Your manuscript is now being handed over to our production team.

Kind regards,

on behalf of

Dr. Lakshminarayana Chekuri

Academic Editor

PLOS One